# A/B Testing in Dense Large-Scale Networks: Design and Inference

**Preetam Nandy, Kinjal Basu, Shaunak Chatterjee, Ye Tu**
LinkedIn Corporation
Mountain View, CA 94083
{ pnandy, kbasu, shchatte, ytu } @linkedin.com

## Abstract

Design of experiments and estimation of treatment effects in large-scale networks, in the presence of strong interference, is a challenging and important problem. Most existing methods' performance deteriorates as the density of the network increases. In this paper, we present a novel strategy for accurately estimating the causal effects of a class of treatments in a dense large-scale network. First, we design an approximate randomized controlled experiment by solving an optimization problem to allocate treatments in the presence of competition among neighboring nodes. Then we apply an importance sampling adjustment to correct for any leftover bias (from the approximation) in estimating average treatment effects. We provide theoretical guarantees, verify robustness in a simulation study, and validate the scalability and usefulness of our procedure in a real-world experiment on a large social network.

## 1 Introduction

Measuring the effect of treatment variants is a fundamental problem in several fields of study [2, 22, 14]. A/B testing is a commonly used method in the internet industry [18, 19, 25, 29] and beyond, wherein randomized experiments are run with two or more such variants. Traditional A/B testing depends on the key assumption that the effect of treatment on an experiment unit is independent of the treatment allocation to other experiment units – commonly called "Stable Unit Treatment Value Assumption" (SUTVA) [22]. However, in many important network settings, this assumption is violated due to various forms of interference [3, 4, 13, 16, 17, 23, 26, 28].

We focus on one such class of problems – a marketplace of commodity producers and consumers. An example is a social media platform, where the commodity is content, and the return utility is feedback. Examples include Facebook, Instagram, LinkedIn, Twitter, etc. Other marketplace problems differ primarily on the commodity and return utility in question. Rides are commodities in Uber and Lyft, retail items are commodities in marketplaces like Amazon and eBay, and money is the common return utility. Note that we do not make any assumptions on how the utility affects a producer's behavior (i.e., the incentive function).

In the marketplace setting, we define a treatment class represented by **a modification of the edge-level probability $p_{ij}$ of showing a commodity of a producer $i$ to a consumer $j$**. This abstraction represents a very general class of treatments since any redistribution or shift in producer and consumer exposure can be expressed via the change in the edge-level probabilities. In content marketplaces, such treatments can reshape the content creator's exposure to their audience, whereas, in a transportation marketplace, it can reshape the likelihood of specific drivers and riders being matched.

In such marketplace problems, SUTVA is violated because of interference from the network. For instance, the effect of treatment on a producer (the experimental unit) is influenced by the allocation and effect of the treatment on all potential consumers of that producer, which in turn depend on each

of their producers' experiences (and hence, their allocated treatments) – thus leading to competition among producers connected to common consumers. When SUTVA is violated, the effect of treatment can be measured by allocating experimental units as well as their first-degree neighbors (and second-degree neighbors, depending upon the interference function) to the same treatment (or control) [10, 23]. The number of such units for which we can successfully allocate treatments in this fashion (i.e., whose neighborhood is appropriately treated) decreases with increasing density in the graph, resulting in lower statistical power in the measurements.

In this paper, we propose a novel technique named OASIS, "**O**ptimal **A**llocation **S**trategy and **I**mportance **S**ampling Adjustment", that provides a randomized testing framework for large-scale marketplace problems in the presence of interference. The performance of OASIS does not deteriorate (and may improve) as the density of the network increases, and it can be used to obtain high-power measurements. Our approach relies on the existence of an "intervening variable" (a.k.a. mediator) — i.e., the effect of the treatment allocation to an experimental unit's network on the unit itself is fully captured by a sufficient statistic (cf.[1, 8, 20]). For example, the total feedback received by a content producer (or the total money made by a driver) can be an intervening variable. The producers in a marketplace compete with each other to receive feedback, and we attempt to mimic this *competition effect* in the design of experiment. Furthermore, we construct an unbiased estimator of the treatment effect by applying an importance sampling correction. We use simulation studies to compare our method against a graph cluster based baseline. From both the simulation studies and from a real-world experiment on a large social network, we show that our technique **(i) compares favorably against the baseline method and also (ii) works very well for large-scale dense graphs that cannot be decomposed into reasonably isolated clusters.**

This technique can be used in any marketplace problem where the treatment in question can be expressed via the $p_{ij}$ "edge weight" abstraction, and where the key assumptions (outlined as Assumptions 1 and 2 in later sections) are satisfied. In many applications, the treatment needs to be applied over a time window during which $p_{ij}$ can change periodically (e.g., consumers' favored producers may change over time) and this dynamism can be handled by updating the experiment design periodically (see the real-world experiment in Section 5).

The rest of the paper is organized as follows. Section 2 formulates the problem and discusses the setup in detail. The key aspects of how we design an experiment via an optimization formulation are described in Section 3. We propose the OASIS estimator and provide theoretical guarantees in Section 4. In Section 5, we provide some empirical results both from an elaborate simulation study and a real-world experiment on a large social network graph, before concluding with a discussion in Section 6. We end this section with a selective literature review.

**Related Work:** Recently there has been a lot of focus on A/B testing in the presence of interference, especially on large real-world networks, with several proposed approaches. This is due to the ever-growing popularity of several marketplace products mentioned previously, and critical applications in those products violating SUTVA in randomized testing. Under the assumption that the interference between units has some known form, [3] propose inverse probability weighting based methods to estimate effects defined in terms of "effective treatments" and refines the estimator via covariance adjustment, and [26] propose to estimate causal effects under partial interference (i.e., when $k$ neighbors are in treatment) using a sequential design. Unlike [3, 26], we aim to estimate the full treatment effect, i.e., all neighbors are in treatment vs. no neighbor is in treatment, for which the sequential design strategy is expected to break down ([26] used $k = 4$ in the experiments).

Network bucket testing [4, 17] looks at a form of interference where a new feature will take effect only if some minimal number of a treated user's neighbors are also in treatment. The key idea, proposed in [4] is to use a walk-based sampling method to generate a core set of users who are internally well-connected while being approximately representative of the population. [17] generalizes this further and provides variance bounds for various core set generation functions. Graph cluster randomization is a related approach introduced in [28]. In this work, a notion of "network exposure" is defined, where a unit is "network exposed" to treatment if its behavior under a particular assignment is the same as its behavior if everyone were assigned to treatment. For various definitions of network exposure, a clustering approach is proposed, and randomization is done at the cluster level. In a real-world network, it very difficult to obtain a large number of reasonably isolated clusters, resulting in low-powered experiments. To mitigate this issue, [23] proposed ego-network randomization, where a cluster is comprised of an "ego" (a focal individual), and her "alters" (the individuals she is

immediately connected to). Our class of treatment definitions can be viewed as a specific instance of network exposure. However, **our randomization is still at the node level and allows us to obtain much higher-confidence measurements with much smaller treatment exposure**.

In the marketplace setting, [30] introduced the bipartite randomized experiment framework, where two distinct groups of units are linked together through a bipartite graph. The treatment is assigned to the first group, called diversion units, and the response is measured on the second group, called outcome units. Under a linear treatment exposure model, [21] proposed a clustering-based approach for estimating the exposure-response function, whereas [9] proposed a generalized-propensity-score [15] based estimator. A treatment exposure model relates the responses of the outcome units to a measure of treatment exposure they receive (cf. Definition 1). The measure of treatment exposure is considered to be a linear function of the treatment assignments of the diversion units in a linear exposure model (cf. Assumption 1). Our framework differs from the bipartite randomized experiment framework in the following two major aspects. First, we consider general directed graphs where each experiment unit can be a diversion unit, an outcome unit, or both. Second, we consider a setting with limited exposure to each outcome unit, leading to a competition among the treatment units for being exposed to an outcome unit. The limited exposure is a common feature of many marketplace settings, where the consumers (e.g., buyers in a commodity marketplace or viewers in a content marketplace) can only view a limited number of items. Finally, we note that our approach does not require extrapolation of the estimated exposure-response function for estimating the average treatment effect, which caused an additional overhead in approaches proposed by [9, 21].

Two other related classical lines of work in estimating causal effect is the instrumental variable approach [1, 8] and the mediation or intervening variable effect estimation [20]. Both approaches make key assumptions about the how the causal effect flows through specific known variables to affect the outcome, and is similar to the assumption we make in order to facilitate replicating a completely treated universe with a much smaller exposure.

## 2   Problem Setup

We describe the problem in a content marketplace setup, where each member is a node in a directed graph (or network) $\mathcal{G} = (\Omega, E)$, and each member is either a content-consumer, a content-producer or both. If $i \to j$, then producer $i$ is a parent of $j$ and consumer $j$ is a child of $i$. We denote the set of all parents of $j$ by $Pa(j)$ and the set of all children of $i$ by $Ch(i)$. When a consumer $j$ visits the marketplace she views a content produced by one of her parents $i$ with probability $p_{ij}$, where $\sum_{i \in Pa(j)} p_{ij} = 1$. We denote the probabilities in the existing system (i.e., prior to any intervention) by $p_{ij}^{base}$ and an intervention (or a treatment) on member $j$ replaces her consumer-side experience $\{p_{ij}^{base} : i \in Pa(j)\}$ by $\{p_{ij}^{(r)} : i \in Pa(j)\}$ satisfying $\sum_{i \in Pa(j)} p_{ij}^{(r)} = 1$.

Let $T_E^{(r)} = \{p_{ij}^{(r)} : (i,j) \in E\}$ be an intervention over the entire population and $\tau_r = \mathbb{E}[Y(T_E^{(r)})]$ be the expected response of the population under $T_E^{(r)}$. We assume that the responses of members $i$ and $j$, denoted by $Y_i(T_E^{(r)})$ and $Y_j(T_E^{(r)})$ respectively, are independently distributed for a given $T_E^{(r)}$. We consider $m$ treatments $T_E^{(1)}, \dots T_E^{(m)}$ and a control environment $T_E^{(0)} = \{p_{ij}^{base} : (i,j) \in E\}$. Our goal is to estimate the average treatment effects $\tau_1 - \tau_0, \dots, \tau_m - \tau_0$ in a marketplace setup (see Figure 1b and 1a) defined below.

*Definition* 1. A directed network $\mathcal{G} = (\Omega, E)$ of members is said to be a marketplace if member $i$'s response $Y_i(T_E^{(r)})$ depends on the treatment $T_E^{(r)}$ through

1. **Consumer-side experience**: the exposure of producers to consumer $i$: $\{p_{ki}^{(r)} : k \in Pa(i)\}$, and

2. **Producer-side experience**: the exposure of producer $i$ to consumers: $\{p_{ij}^{(r)} : j \in Ch(i)\}$.

In a marketplace setup, $T_E^{(r)}$ not only has a direct effect on $Y_i(T_E^{(r)})$ through the modification of the existing consumer-side experience $\{p_{ri}^{base} : r \in Pa(i)\}$ (i.e., what contents have been shown to $i$), but also has an indirect effect on $Y_i(T_E^{(r)})$ through the modification of the producer-side

experience $\{p_{ij}^{base} : j \in Ch(i)\}$ (i.e., other members liking, sharing or commenting on the contents produced by $i$). The latter violates the SUTVA assumption and is known as the network effect. Further, each producer competes with its second-degree neighbors, to obtain higher $p_{ij}^{(r)}$ values as $\sum_{i \in Pa(j)} p_{ij}^{(r)} = 1$ for all $j$. This was referred as the competition effect in Section 1. In Figure 1a, the producer-side experience of member 1 depends on the feedback received from consumers 2, 3 and 5, while the consumer-side depends on the incoming edges from producers 4 and 5. Due to this fact, there is a competition between member 4 and member 5 for getting the feedback from consumer 1.

Recall that a classical A/B testing setting would consider randomly selected non-overlapping subsets $\Omega_0, \ldots, \Omega_m$ of $\Omega$ and assign treatment $\{p_{ki}^{(r)} : k \in Pa(i)\}$ to each $i \in \Omega_r$ and estimate $\tau_r - \tau_0$ by

$$\hat{\tau}_r - \hat{\tau}_0 = \frac{1}{|\Omega_r|} \sum_{i \in \Omega_r} Y_i^{(exp)} - \frac{1}{|\Omega_0|} \sum_{i \in \Omega_0} Y_i^{(exp)}, \tag{1}$$

where $Y_i^{(exp)}$ is the response of member $i$ under the aforementioned experimental design. It is easy to see that the estimator defined in (1) is not unbiased as the producer-side experience of member $i \in \Omega_r$ is not the same in the classical A/B testing design and under $T_E^{(r)}$, since all $j \in Ch(i)$ is not guaranteed to be in $\Omega_r$. The bias of the estimator in (1) is known as the "spillover effect" in the literature. A commonly used technique for mitigating the spillover effect is to choose $\Omega_0, \ldots, \Omega_m$ as disjoint from each other as possible. This is called the cluster-based approach. Even for moderately dense networks, a cluster-based method does not provide a reasonable solution, since it either induces a large bias by ignoring inter-cluster edges and/or suffers from the loss of power by considering a small sample consisting of nearly perfect clusters. In this paper, we propose a complementary approach under the following assumption.

**Assumption 1.** *The producer-side experience of member $i$ depends on the treatment condition $T_E^{(r)}$ only through a weighted total exposure $Z_i(T_E^{(r)}) = \sum_{j \in Ch(i)} \alpha_{ij} p_{ij}^{(r)}$ for some constants $\alpha_{ij}$'s representing the strength of the relationship between producer $i$ and consumer $j$ (see Figure 1b).*

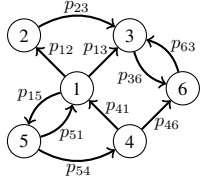

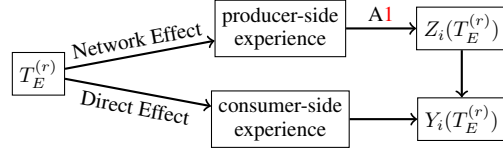

**(1a)** A producer-consumer network.　　　　**(1b)** A graphical representation of Assumption 1.

In the content marketplace setup, Assumption 1 corresponds to the situation where the producer-side experience of member $i$ depends only on an aggregated feedback to member $i$'s content. Note that as the potential number of consumers for member $i$'s content increases, it becomes more likely that the total feedback determines the producer-side experience. Thus, Assumption 1 becomes more reasonable, as the density of the network increases.

In Section 3, we construct an experiment $T_E^*$ where we exactly match $p_{ki}^*$ with $p_{ki}^{(r)}$ for all $i \in \Omega_r$ and $k \in Pa(i)$ (as in the classical A/B testing design), and attempt to match $Z_i(T_E^*)$ with $Z_i(T_E^{(r)})$ as much as possible by solving a joint optimization problem for all $i \in \Omega_r$ and $r \in [m]$ (where $[m] = 0, \ldots, m$). In practice, it is reasonable to assume that $\alpha_{ij}$'s are unknown or approximately known. In Section 3.1, we show the robustness of our estimator under the violation of the assumption that $\alpha_{ij}$'s are known. Finally, we propose an importance sampling based adjustment to correct for experimental bias and describe the OASIS estimator.

## 3 Design of Experiment

We present our design in Algorithm 1 and an example in Figure 2. Let $\Omega_r$, $\Lambda_r$, and $C'$ be disjoint subsets of $\Omega$ (as in Algorithm 1), and let $\Omega' = \cup_{r=0}^m \Omega_r$ and $\Lambda' = \cup_{r=0}^m \Lambda_r$. The three disjoint subsets represent the following:

1. $\Omega_r$: This is the population which will be used to measure $Y(T_E^{(r)})$. $|\Omega_r|$ determines the statistical power of the measurement. We allocate the exact consumer-side experience for each member $i \in \Omega_r$. The producer side experience is approximately matched by an optimization formulation (see below).

2. $\Lambda_r$: This is a shadow population for which we also allocate the exact consumer-side experience. As shown in Figure 2, since $\Omega_r$ matches both the consumer-side and producer-side experience while $\Lambda_r$ matches only the consumer side experience, we can compare $\Omega_r$ and $\Lambda_r$ to detect the presence of network effect.

3. $C'$: Members whose incoming edges from $\Omega'$ are assigned weights to match the producer experience for members in $\Omega'$.

In order to achieve the above objectives, we assign edges in the following manner:

1. $p_{ij}^{(r)}$ to all incoming edges to members in $\Omega_r \cup \Lambda_r$, hence exactly matching their consumer side experience.

2. a new treatment $p_{ij}^*$ to all incoming edges to members in $C'$, to approximately match the producer side experience of members in $\Omega'$, and

3. $p_{ij}^{base}$ to all incoming edges to members not in $\Omega_r \cup \Lambda_r \cup C'$, to control the overall risk.

We solve an optimization problem to define the new treatment $p_{ij}^*$ for all edges from $\Omega'$ to $C'$ and line (6) of Algorithm 1 defines $p_{ij}^*$ of all other incoming edges to $C'$ such that $\sum_{i \in Pa(j)} p_{ij}^* = 1$ for all $j \in C'$. The selection probability $q$ in Algorithm 1 controls the number of members $|C'|$ that are receiving the new treatment.

---

**Algorithm 1** Optimal Allocation Strategy (OAS)

---

1: Randomly select disjoint subsets $\Omega_0, \Omega_1, \ldots, \Omega_m$ of $\Omega$ and define $\Omega' = \cup_{r=0}^m \Omega_r$;
2: Randomly select disjoint subsets $\Lambda_0, \Lambda_1, \ldots, \Lambda_m$ of $\Omega \setminus \Omega'$ and define $\Lambda' = \cup_{r=0}^m \Lambda_r$;
3: Set $p_{ij}^* = p_{ij}^{(r)}$ for all $j \in \Omega_r \cup \Lambda_r, r = 0, \ldots, m, i \in Pa(j)$;
4: Construct a subset of children $C'$ by selecting each member in $\cup_{i \in \Omega'} Ch(i) \setminus (\Omega' \cup \Lambda')$ with probability $q$;
5: Obtain $\{p_{ij}^* : j \in C', i \in Pa(j) \cap \Omega'\}$ by solving the optimization problem given in (2);
6: Set $p_{ij}^* = p_{ij}^{base} \times \frac{1 - \sum_{k \in Pa(j) \cap \Omega'} p_{kj}^*}{1 - \sum_{k \in Pa(j) \cap \Omega'} p_{kj}^{base}}$ for $j \in C'$ and $i \in Pa(j) \setminus \Omega'$;
7: For all other $(i, j) \in E$, set $p_{ij}^* = p_{ij}^{base}$;

---

The sets $\Lambda_r$ will also be used in the importance sampling step described in Section 4. Thus, the existence of $\Lambda_r$ increases the accuracy of our final estimator without increasing the time complexity of the optimization problem. Further, as mentioned above, $\Omega_r$ and $\Lambda_r$ can be compared to detect the presence of the network effect in a real-world experiment.

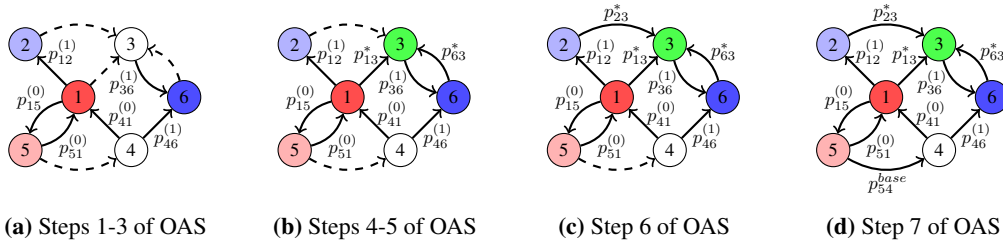

(a) Steps 1-3 of OAS      (b) Steps 4-5 of OAS      (c) Step 6 of OAS      (d) Step 7 of OAS

**Figure 2:** Consider Algorithm 1 with the network in Figure 1a. Let $\Omega_0 = \{1\}, \Omega_1 = \{6\}, \Lambda_0 = \{5\}, \Lambda_1 = \{2\}$ and $C' = \{3\}$. In (a), we assign the color red or blue to a node $j$ with consumer-side experience $\{p_{ij}^{(0)} : i \in Pa(j)\}$ or $\{p_{ij}^{(1)} : i \in Pa(j)\}$ respectively. In (b) we assume $\alpha_{ij} = 1$ for all $(i, j) \in E$ and obtain $\{p_{13}^*, p_{63}^*\}$ by minimizing $(p_{12}^{(0)} + p_{13}^{(0)} - p_{12}^{(1)} - p_{13})^2 + (p_{63}^{(1)} - p_{63})^2$. Next, we assign $p_{23}^* = p_{23}^{base} \times \frac{1 - (p_{13}^* + p_{63}^*)}{1 - (p_{13}^{base} + p_{63}^{base})}$ in (c). Finally, we assign $p_{54}^{base}$ to the edge $5 \to 4$ in (d).

## 3.1 Optimization Problem

Let $E^* \subseteq E$ be the set of all edges from $\Omega'$ to $C'$. Our goal is to obtain $\{p_{ij}^* : (i,j) \in E^*\}$ such that $J(E^*) = \sum_{r=0}^m \sum_{i \in \Omega_r} (Z_i(T_E^{(r)}) - Z_i(T_E^*))^2$ is minimized. Recall that $Z_i(T_E^{(r)}) = \sum_{j \in Ch(i)} \alpha_{ij} p_{ij}^{(r)}$. By Algorithm 1, we first assign all edge weights that won't be determined by the optimization. This includes assigning $p_{ij}^{(s)}$ to all edges $E_{i,s}$ from $i$ to members in $\Omega_s \cup \Lambda_s$ and assign $p_{ij}^{base}$ to all edges $E_{i,base}$ from $i$ to members not in $\Omega_s \cup \Lambda_s \cup C'$. To capture this already assigned exposure of a member $i \in \Omega'$, we define

$$Z_i'(T_E^*) = \sum_{s=0}^m \sum_{(i,j) \in E_{i,s}} \alpha_{ij} p_{ij}^{(s)} + \sum_{(i,j) \in E_{i,base}} \alpha_{ij} p_{ij}^{base}.$$

Next, we denote the set of all edges from $i$ to $C'$ by $E_{i,C'}$. All outgoing edges from $i$ is the union of $\{E_{i,s} : s = 0, \ldots, m\}$, $E_{i,base}$ and $E_{i,C'}$ and we adjust the exposure level corresponding to $E_{i,C'}$ to minimize $(Z_i(T_E^{(r)}) - Z_i(T_E^*))^2$ simultaneously for all $i \in \Omega_r$, $r \in \{0, \ldots, m\}$. Also, note that $\cup_{i \in \Omega'} E_{i,C'} = E^*$. Therefore, we find $\{p_{ij}^* : (i,j) \in E^*\}$ by solving

$$
\begin{aligned}
\underset{p_{ij}}{\text{Minimize}} \quad & \sum_{r=0}^m \sum_{i \in \Omega_r} \left( Z_i(T_E^{(r)}) - Z_i'(T_E^*) - \sum_{(i,j) \in E_{i,C'}} \alpha_{ij} p_{ij} \right)^2 \\
\text{subject to} \quad & \sum_{i \in Pa(j) \cap \Omega'} p_{ij} \leq 1, \\
& p_{ij} \geq 0.
\end{aligned}
\tag{2}
$$

**Risk Control**: The risk of an experiment is defined as the total negative impact of an online experiment on members' experience. The risk of OAS tend to be higher than a classical A/B testing design due the modification in the consumer-side experience of members in $C'$. We can control the risk by restricting the size of $C'$, and by imposing additional constraints to the optimization problem to control the deviation of $p_{ij}^*$ from $p_{ij}^{base}$ (see Appendix A.1).

**Scaling the Optimization**: The quadratic programming (QP) defined in (2) does not scale well with the number of edges $n := |E^*|$. Even in moderately sized experiments in social network graphs, we can expect $n$ to range in billions, making it almost impossible to solve the QP. We propose an iterative approximation that iterates over $K$ overlapping sub-problems of roughly equal size $n_{sub}$, where each sub-problem respects the constraints of the full optimization and the solution of each sub-problem can improve the potential solution obtained in the previous step. By following this strategy, we achieve an $\mathcal{O}(K \times n_{sub}^3)$ time complexity, which is much better than the $\mathcal{O}(n^3)$ time complexity of QP when $K^{1/3} \times n_{sub} << n$. We use the Operator Splitting method to solve each QP [5, 6, 24]. The detailed algorithm and simulation results are provided in Appendix A.2.

**Robustness**: In the Appendix A.3, we show the robustness of the OAS design in the sense that we can work with partially known $\alpha_{ij}$'s (see Assumption 1) or assume them to be equal to one without inducing any bias, as long as certain assumptions of independence are satisfied. More precisely, if we design a treatment $T_E^*$ and a member $i$ such that $p_{ki}^* = p_{ki}^{(r)}$ for all $k \in Pa(i)$ and

$$\sum_{j \in Ch(i)} \beta_{ij} p_{ij}^* = \sum_{j \in Ch(i)} \beta_{ij} p_{ij}^{(r)},$$

then $\mathbb{E}[Y_i(T_E^*)] = \mathbb{E}[Y_i(T_E^{(r)})]$ holds under much less strict condition than the equality of $\beta_{ij}$'s and $\alpha_{ij}$'s. In our simulation study in Section 5, consider further violations of Assumption 1, where $Z_i(T_E^{(r)})$'s are chosen to be non-linear functions of $p_{ij}^{(r)}$'s.

## 4 Post-experiment Bias Correction

*Importance Sampling Adjustment*: Under the following assumption, we propose an unbiased estimator of the average treatment effect $\tau_r - \tau_0$ under a weaker requirement than a "perfect" design of experiment that facilitates the unbiased estimation using (1).

**Assumption 2.** *For any treatment $T_E^{(r)}$, the producer-side experience of member $i$ depends on $T_E^{(r)}$ only through the total exposure $Z_i(T_E^{(r)}) = \sum_{j \in Ch(i)} Z_{ij}(T_E^{(r)})$ for some observable i.i.d. random variables $Z_{ij}$'s. We assume that $Z_i(T_E^{(r)})$ and $\{p_{ki}^{(r)} : k \in Pa(i)\}$ are independently distributed.*

The first part of Assumption 2 is weaker than Assumption 1 where we additionally assume $Z_{ij}(T_E^{(r)}) = \alpha_{ij} p_{ij}^{(r)}$. Also, note that the existence of an intermediate variable or mediator $Z_i(T_E^{(r)})$ need not be unique. For example, in the content marketplace setup, $Z_{ij}(T_E^{(r)})$ can be a function of $p_{ij}^{(r)}$, as well as $Z_{ij}(T_E^{(r)})$ can be a more downstream variable representing the feedback of member $j$ on the content of member $i$. In the latter case, the second part of Assumption 2 corresponds to the assumption that the feedback on member $i$'s content is independent of the contents shown to member $i$, which is a reasonable assumption for most real-world recommendation systems.

**Theorem 1.** *Let $f_r$ denote the density of $Z_i(T_E^{(r)})$, and let $f_r^*$ denote the density of $Z_i(T_E^*)$ conditionally on $i \in \Omega_r$, where $T_E^*$ is the output of Algorithm 1. We define*

$$w_i = \frac{f_r(Z_i(T_E^*))}{f_r^*(Z_i(T_E^*))}, \ \ and \ \ \hat{\tau}_r = \frac{1}{|\Omega_r|} \sum_{i \in \Omega_r} w_i Y_i(T_E^*)$$

*Then under Assumption 2, we have $\mathbb{E}[\hat{\tau}_r] = \tau_r, \forall \, r \in \{0, \dots, m\}$.*

The correctness of Theorem 1 does not rely on Assumption 1 and the experimental design. However, from a practical standpoint, if we (approximately) know the dependency of $Z_i(T_E)$, we should use it to do the matching while designing the experiment. It is crucial for reducing the variance of the importance sampling weights as well as the corresponding estimator. Further, we recommend to use the self-normalized importance sampling $\frac{1}{\sum w_i} \sum Y_i w_i$, as it is known to be more stable in practice.

The densities $f_r^*$ and $f_r$ in Theorem 1 can be estimated using the data $\{Z_i(T_E^*) : i \in \Omega_r\}$ and the data $\{Z_{ij}(T_E^{(r)}) : i \in \Omega_r, \ j \in Ch(i) \cap (\Omega_r \cup \Lambda_r)\}$ respectively, where $\Omega_r$ and $\Lambda_r$ are as in Algorithm 1. For further the details, please see Appendix A.5.

*OASIS*: We conclude this section by gluing all the pieces together in Algorithm 2, called Optimal Allocation Strategy and Importance Sampling Adjustment (OASIS). Under certain assumptions, an asymptotically correct $(1 - \alpha)100\%$ confidence interval for $\tau_r$ is given by $\hat{\tau}_r \pm \Phi^{-1}(1 - \alpha/2) \times \hat{\sigma}_r$, where $\Phi$ is the CDF of the standard normal distribution, where $\hat{\sigma}_r$ is a bootstrap variance estimator [11] (see Appendix A.6). The result follows from the consistency of density estimation, the central limit theorem and the consistency of the bootstrap variance estimator. Note that the estimator in Theorem 1 is an empirical average when the densities are known, implying the consistency of the bootstrap variance estimators under mild assumptions. When the densities are unknown, the consistency of the bootstrap variance relies on the consistency of the density estimation (which holds under some regularity conditions).

---

**Algorithm 2** OASIS

---
1: Obtain $T_E^*$ by applying Algorithm 1;
2: Run experiment $T_E^*$ to collect data;
3: Obtain estimated densities $\hat{f}_r^*$ and $\hat{f}_r$ of $\{Z_i(T_E^*) : i \in \Omega_r\}$ and $\{Z_i(T_E^{(r)}) : i \in \Omega_r\}$ using the aforementioned technique, for $r = 0, \dots, m$;
4: Obtain $\hat{\tau}_r = \frac{1}{|\Omega_r|} \sum_{i \in \Omega_r} Y_i(T_E^*) \frac{\hat{f}_r(Z_i(T_E^*))}{\hat{f}_r^*(Z_i(T_E^*))}$ for $r = 0, \dots, m$;
5: Return $\hat{\tau}_1 - \hat{\tau}_0, \ \dots, \ \hat{\tau}_m - \hat{\tau}_0$.

---

## 5 Experiments

*Simulation Study*: We compare OASIS with a graph-cluster randomization method that randomly assigns treatment (or control) to all members in a cluster and uses (1) for estimating treatment effects. The cluster-based (CB) method yields an unbiased estimator when the clusters are disjoint, and the performance of CB is expected to deteriorate as the number of inter-cluster edges increase. We

generate graphs with 50000 nodes from some well studied graphical models [7, 12] with 5 equal-sized clusters and varying density (i.e. average degree) and varying inter-cluster edge proportion (see Appendix A.7.1). We directly use these clusters (instead of estimating them) for graph-cluster randomization to avoid the dependency on the performance of a graph clustering technique (see Appendix A.7.2). While the chosen network topology (i.e. the presence of clusters) and the use of the known clusters is somewhat favorable to CB, the presence of clusters does not provide any advantage to the OASIS design and estimates.

We generate data under moderate violations of Assumption 1. More precisely, we set $Z_i(T_E^{(r)})$ as $Z_i(\delta, T_E^{(r)}) = \sum_{j \in Ch(i)} \alpha_{ij} \, (p_{ij})^\delta$ for $r \in \{0, 1\}$, $\delta \in \{1/4,\ 1/2,\ 1\}$. **Observe that the strength of the network effect $Z_i^{(r)}(\delta, T_E^{(r)})$ decreases as $\delta$ increases**. We choose $\alpha_{ij} = 1$ for constructing $T_E^*$ using Algorithm 1. The details on the graph and data generation and the treatment definitions are given in the Appendix.

| $\bar{d}_\mathcal{G}$ | $\tau_0$ | $\tau_1 - \tau_0$ | OASIS | CB | $\tau_0$ | $\tau_1 - \tau_0$ | OASIS | CB | $\tau_0$ | $\tau_1 - \tau_0$ | OASIS | CB |
|---|---|---|---|---|---|---|---|---|---|---|---|---|
| 21 | 9.81 | 0.042 | 0.958 | 0.974 | 9.10 | 0.165 | 0.968 | 0.976 | 6.76 | 0.262 | 0.928 | 0.946 |
| 55 | 9.73 | 0.064 | 0.950 | 0.948 | 8.32 | 0.233 | 0.950 | 0.946 | 5.84 | 0.137 | 0.954 | 0.938 |
| 120 | 9.61 | 0.089 | 0.956 | 0.914 | 7.55 | 0.234 | 0.938 | 0.592 | 5.50 | 0.078 | 0.932 | 0.880 |
| | **(a)** $\delta = 1/4$ | | | | **(b)** $\delta = 1/2$ | | | | **(c)** $\delta = 1$ | | | |

**Table 1:** Empirical coverage probability for 95% confidence intervals, computed based on $n = 500$ experiments. Higher coverage corresponds to higher accuracy in measurement. $\bar{d}_\mathcal{G}$ denotes the average degree of the graph.

Table 1 shows the empirical coverage probabilities defined as the proportion of times $\tau_1 - \tau_0$ lies in the corresponding 95% confidence interval. Ideally, the empirical coverage probability should match $0.95$, a lower value would indicate a higher Type-1 error and a higher value would indicate a higher Type-2 error. The setting $\bar{d}_\mathcal{G}$ is the most unfavorable setting for CB with the proportion of inter-cluster edges, and among all settings with $\bar{d}_\mathcal{G}$, the setting with $\delta = 1/2$ has the largest treatment effect. We suspect that CB failed to capture a significant portion of the network effect under these extreme conditions.

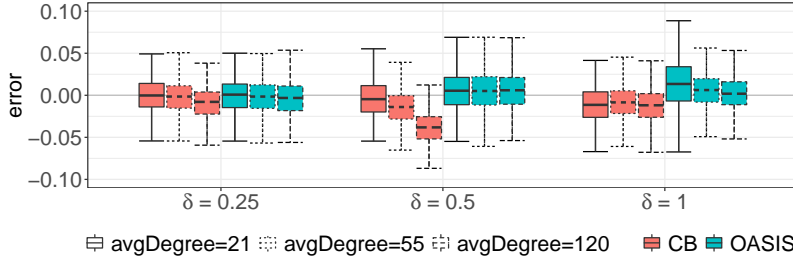

**Figure 3:** Box-plots of the estimation errors under different settings corresponding to $\delta \in \{1/4, 1/2, 1\}$ and $\text{avgDegree} = \bar{d}_\mathcal{G} \in \{21, 55, 120\}$ based on 500 repeats.

We compare the errors of estimating $\tau_1 - \tau_0$ with the OASIS and the cluster-based estimators in Figure 3. We observe two important patterns:

1. The performance of the cluster-based method deteriorates as we move from a sparse and low impurity setting to a dense and high impurity setting, while the performance of OASIS remains the same or gets better as the density of the network increases. Note that with increasing density in the graph, the experiment design part of OASIS gets more flexibility to redistribute the exposure optimally, and the post-experiment adjustment part gets a larger sample size for density estimation (see Section A.5 in the supplement).

2. Performance is better, and the difference between OASIS and the cluster-based method is higher when the treatment effect is larger (e.g., the middle set of experiments, where $\tau_1 - \tau_0$ is large).

***Real World Experiments***: We demonstrate an application of our method on the LinkedIn newsfeed, a content recommender system with 675M+ members. A content recommender system is often optimized for content consumers' experience. However, there has been a growing interest in developing

recommender systems that additionally enhance content producers' experience [27]. It is easy to see that the classical A/B testing framework won't be able to capture the enhancement in the producers' experience. We apply OASIS for A/B testing a new recommender system $M^{(1)}$ and a baseline model $M^{(0)}$. For details on the setup, please see Appendix A.8. To estimate the $p_{ij}^{(0)}$ and $p_{ij}^{(1)}$ values, we ran a preliminary experiment for one week where we assigned $M^{(0)}$ and $M^{(1)}$ to members in $\Omega_0 \cup \Lambda_0 \cup \Gamma$ and $\Omega_1 \cup \Lambda_1$ respectively. The assignment of $M^{(r)}$ to member $i$ means that the LinkedIn newsfeed of member $i$ generated by $M^{(r)}$ as follows. An item-generator model creates a list of items for each member $i$, then each $item$ in the list receives a score $M^{(r)}(item, i)$, and finally the items are shown to member $i$ ranked by their scores.

Let's denote the sum of the scores of all items of producer $k$ that are shown to member $i$ by $S^{(r)}(k, i)$. Then,

$$p_{ij}^{(r)} := \frac{S^{(r)}(i, j)}{\sum_{k \in Pa(j)} S^{(r)}(k, j)}.$$

Moreover, we define

$$Z_i(T_E^{(r)}) := \frac{|\Omega|}{|\Omega_r \cup \Lambda_r|} \sum_{j \in Ch(i) \cap (\Omega_r \cup \Lambda_r)} p_{ij}^{(r)} \text{ for } r = 0, 1.$$

We assume $\alpha_{ij} = 1$ and we obtain $\{p_{ij}^* : (i, j) \in E^*\}$ by minimizing $J(E^*)$ defined in Section 3, where $E^*$ is the set of all edges from $\Omega'$ to $C'$. We started running the main experiment by modifying the scores generated by $M^{(0)}$ with a multiplicative boost factors $b_{ij}$'s (see Appendix A.8.1) such that

$$\frac{b_{ij} S^{(0)}(i, j)}{\sum_{k \in Pa(j)} b_{kj} S^{(0)}(k, j)} = p_{ij}^*, \quad \text{for all incoming edges to members in } C'. \quad (3)$$

We ran the main experiment for three weeks while updating $p_{ij}^*$'s and $b_{ij}$'s periodically (twice a week) to incorporate the dynamic nature of the underlying network. The OASIS estimates (based on the main experiment) showed significant impact in the following metrics that are related to the producer side experience of a member:

(i) *Daily Unique Contributors*: Number of unique members in a day who contribute on (i.e., either like, comment or share) items on the newsfeed, showed a lift of $+\mathbf{0.52\%}$ with a $p$-value of $0.041$.

(ii) *Members in Active Community*: Number of members who contribute to or receive contributions from at least $5$ members, showed a lift of $+\mathbf{1.14\%}$ with a $p$-value of $0.016$.

(iii) *Feed Interaction received Uniques*: Number of unique members who receive at least one contribution, showed a lift of $+\mathbf{1.34\%}$ with a $p$-value of $0.001$.

The above results compares the responses of members in $\Omega_1$ with the members in $\Omega_0$. A similar comparison for $\Lambda_1$ with the members in $\Lambda_0$ turns out to be statistically insignificant for these metrics, indicating the presence of network effect.

## 6 Discussion

We have presented a two-step method, called OASIS, for estimating the average treatment effect for a class of treatments represented by modifications of edge weights in networks with interference. First, we design an experiment by optimally allocating a "modified" treatment exposure to a set of randomly selected consumer-producer pairs by solving a large-scale quadratic program. Secondly, we apply an importance sampling correction for estimating the average treatment effect, which corrects for the design bias induced by the violations of assumptions and/or the restrictions applied for risk control. The main advantage of OASIS compared to cluster-based methods is that it tends to perform better for dense networks that cannot be decomposed into reasonably isolated clusters, while the cluster-based methods would have an advantage over OASIS for sparse networks. An interesting future work could be to combine a cluster-based approach with OASIS in order to gain additional robustness and efficiency. Finally, note that we can handle dynamic networks where the edge weights $p_{ij}$ change over the experiment time window by updating the OASIS design periodically. In some cases, $p_{ij}$ may change very quickly with time or depend on the current context (e.g., drivers and riders are matched based on current proximity). Therefore, considering the distribution of $p_{ij}$ (instead of a point estimate) for designing the OASIS treatment allocation can be another interesting future work.

## Broader Impact

Online A/B testing plays the most crucial role in product development in the internet industry by deciding which recommender system is an optimal choice. The optimality criterion often focuses on the experience of the direct users of the recommender system, e.g., the viewers of a social media newsfeed or the recruiters searching for potential candidates. This choice can have a potentially negative impact on the people indirectly affected by these recommendation choices. For example, the less popular content creators or a certain group of jobseekers might be getting less exposure than the already popular creators or candidates, leading to a "rich get richer" ecosystem. There has been a growing interest in enhancing the experience of the people that are indirectly affected. However, a significant obstacle to achieving this goal is that these indirect effects are challenging to measure in an online A/B testing experiment. In this paper, we take a step toward solving this problem, which can potentially lead to product developments focusing on the holistic improvement of the underlying ecosystem.

## Acknowledgment

We would sincerely like to thank Guillaume Saint-Jacques, Iavor Bojinov, Ya Xu, Lingjie Weng, Shipeng Yu, Hema Raghavan, Romer Rosales and Deepak Agarwal for their support and detailed insightful feedback during the development of this system. We would also like to thank the anonymous reviewers for their helpful comments which has significantly improved the paper.

Finally, none of the authors received any third-party funding for this submission and there is no competing interest other than LinkedIn Corporation to which all authors are affiliated.

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
