[Supplementary Material]

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

under some constraints controlling the risk of overexposure or underexposure of parents of $j \in C'$. We choose $0 \leq R_{\min} \leq 1 \leq R_{\max}$ and $0 \leq S_{\min} \leq 1 \leq S_{\max}$ to control the changes in $\{p_{ij}^{base} : (i,j) \in E^*\}$ and $\{p_{ij}^{base} : j \in C', (i,j) \in E \setminus E^*\}$ respectively by defining $\{p_{ij}^* : (i,j) \in E^*\}$ as a solution of the following quadratic optimization problem with linear constraints:

$$\operatorname*{Min}_{p_{ij}} J(E^*) \text{ such that: } p_{ij}^{base} R_{\min} \leq p_{ij} \leq p_{ij}^{base} R_{\max},$$

$$S_{\min} \leq \frac{1 - \sum_{i \in Pa(j) \cap \Omega'} p_{ij}}{1 - \sum_{i \in Pa(j) \cap \Omega'} p_{ij}^{base}} \leq S_{\max}, 0 \leq \sum_{i \in Pa(j) \cap \Omega'} p_{ij} \leq 1. \tag{4}$$

Note that the last two constraints can be combined as $\ell_j \leq \sum_{i \in Pa(j) \cap \Omega'} p_{ij} \leq u_j$ for $\ell_j = \max\{0, \ 1 - S_{\max}(1 - \sum_{i \in Pa(j) \cap \Omega'} p_{ij}^{base})\}$ and $u_j = \min\{1, \ 1 - S_{\min}(1 - \sum_{i \in Pa(j) \cap \Omega'} p_{ij}^{base})\}$.

## A.2 Scaling the Optimization

We split the set of consumers $C'$ into disjoint sets $\mathcal{S}_k$ of roughly equal sizes, such that $C' = \cup_{k=1}^{K} \mathcal{S}_k$. Since this induces a natural partition in the constraint space of (4), it is easy to see that any candidate solution $\{p_{ij}^{old} : (i,j) \in E^*\}$ of (4) can be improved by updating $\{p_{ij}^{old} : (i,j) \in E^*, j \in \mathcal{S}_k\}$ with the solution of the following optimization problem for all $k \in \{1, \ldots, K\}$. For $i \in \Omega_r$ let

$$\Delta_{ik,r} := Z_i(T_E^{(r)}) - Z_i'(T_E^*) - \sum_{(i,j) \in E_{i,C'}, \ j \notin \mathcal{S}_k} \alpha_{ij} p_{ij}^{old}.$$

$$\operatorname*{Minimize}_{p_{ij}}$$

$$\sum_{r=0}^{m} \sum_{i \in \Omega_r \cap (\cup_{j \in \mathcal{S}_k} Pa(j))} \left( \Delta_{ik,r} - \sum_{(i,j) \in E_{i,C'}, \ j \in \mathcal{S}_k} \alpha_{ij} p_{ij} \right)^2$$

such that

$$R_{\min} \times p_{ij}^{base} \leq p_{ij} \leq R_{\max} \times p_{ij}^{base},$$

$$S_{\min} \leq \frac{1 - \sum_{i \in Pa(j) \cap \Omega'} p_{ij}}{1 - \sum_{i \in Pa(j) \cap \Omega'} p_{ij}^{base}} \leq S_{\max}, \text{ and}$$

$$0 \leq \sum_{i \in Pa(j) \cap \Omega'} p_{ij} \leq 1. \tag{5}$$

Note that (5) and (4) have the same set of constraints for each $j \in \mathcal{S}_k$.

We apply this strategy in Algorithm 3 to solve the overall optimization problem. We start with a feasible candidate $p_{ij}^{old} = p_{ij}^{base}$ and run an iterative scheme to update $p_{ij}$ using (5) as we loop over each $k = \{1, \ldots, K\}$. Once this inner loop completes, we get the full next best $\{p_{ij} : (i,j) \in E^*\}$. We continue the outer loop till convergence. By doing this iterative scheme we are able to solve much larger problems, since size of the each optimization problem $n_k = \sum_{j \in \mathcal{S}_k} |Pa(j) \cap \Omega'|$ is much smaller than $n = \sum_{j \in C'} |Pa(j) \cap \Omega'|$. In fact, the worst-case complexity of the iterative method with $nIter$ outer iterations is given by $\mathcal{O}(maxIter \times \sum_{k=1}^{K} n_k^3)$. This is much better than the $\mathcal{O}(n^3)$ worst-case complexity of (4) when $maxIter = \mathcal{O}(1)$ and $K^{1/3} \times (\max_k n_k) << n$. Although the convergence cannot be guaranteed for $maxIter = \mathcal{O}(1)$, we were able to obtain reasonable solutions

---

**Algorithm 3** Solving for Optimal $p_{ij}$

---

**Input:** Number of Groups $K$, Outer Iteration Limit $maxIter$
1: Create $\mathcal{S}_k$ s.t. $\mathcal{S} = \cup_{k=1}^{K}\mathcal{S}_k$ s.t. $|\mathcal{S}_k| \approx |\mathcal{S}|/K$ for all $k$;
2: Set $p_{ij}^{old} = p_{ij}^{base}$ and $p_{ij}^{*} = p_{ij}^{base}$;
3: **for** $t = 1, \ldots, maxIter$ **do**
4:     **for** $k = 1, \ldots, K$ **do**
5:         Solve the optimization problem as in (5) to obtain $p_{ij}^{*}$ for all $j \in \mathcal{S}_k, i \in \mathbf{Ne}(j) \cap \Omega'$;
6:         Set $p_{ij}^{old} = p_{ij}^{*}$ for all $j \in \mathcal{S}_k, i \in \mathbf{Ne}(j) \cap \Omega'$;
7:     **end for**
8: **end for**
9: Return $\{p_{ij}^{*} : i \in \Omega', j \in \mathcal{S}\}$.

---

in simulations and real applications by using $maxIter = 10$. Finally, note that for a fixed value of $maxIter$, $K$ represents the speed-accuracy trade-off, i.e. the quality of the solution is expected to be better for a smaller value of $K$ but the computational efficiency is expected to be better for a larger value of $K$.

We illustrate the computational efficiency and the accuracy of the iterative algorithm with the following simulation setup. We consider 5 settings with $n = 10000, 20000, 30000, 40000$ and $50000$ variables. For each setting, we generate $n$ producers and $n$ consumers by sampling with replacement from $\Omega = \{1, \ldots, 100\}$. Let $\Omega'$ denote the set of all selected producers and let $E^*$ denote the set of selected producer-consumer pairs. For each producer $i \in \Omega'$, we generate $h_i$ from $Uniform[0, n/100]$ and for each producer-consumer pair $(i, j) \in E^*$, we generate $p_{ij}^{base} \sim Uniform[0, 1]$. We solve the following optimization problem:

$$\underset{p_{ij}}{\text{Minimize}} \sum_{i \in \Omega'} \left( h_i - \sum_{j \in Ch(i)} p_{ij} \right)^2 \text{ such that:}$$

$$0.1 \times p_{ij}^{base} \leq p_{ij} \leq 10 \times p_{ij}^{base}, \quad \text{and} \quad 0.2 \times \sum_{i \in Pa(j)} p_{ij}^{base} \leq \sum_{i \in Pa(j)} p_{ij} \leq 5 \times \sum_{i \in Pa(j)} p_{ij}^{base},$$
$$\tag{6}$$

where the $Pa(i)$ and $Ch(i)$ are defined on the producer-consumer graph $(\Omega, E^*)$.

**Figure 4:** Boxplots of runtimes and the final objective values for solving (6) directly using a QP solver (method: full) and for solving (6) using Algorithm 3 with $K = 10$ and $maxIter = 2$ (method: iterative). We used the Operator Splitting method [24] in R package **rosqp** for solving each subproblem in Algorithm 3 as well as for solving the full QP.

Figure 4 shows that our iterative QP algorithm (Algorithm 3) can outperform the full QP solver in terms of computational efficiency without sacrificing the solution quality (measured by the final objective value).

### A.3 Robustness

**Lemma 1.** *Let $i$ be a randomly chosen member in $\Omega$ and let $\{p_{ki}^{(r)} : k \in Pa(i)\}$ and $\{\alpha_{ij}, p_{ij}^{(r)} : j \in Ch(i)\}$ be well-defined random variables. Assume*

$$Y_i(T_E^{(r)}) = g\left( \sum_{j \in Ch(i)} \alpha_{ij} p_{ij}^{(T_E^{(r)})}, \mathbf{W}_i(T_E^{(r)}) \right) + \epsilon_i,$$

*where $g$ is a differentiable function with respect to the first coordinate, $\mathbf{W}_i(T_E^{(r)}) := \{p_{ki}^{(r)} : k \in Pa(i)\}$, and $\epsilon_i$ does not depend on $T_E^{(r)}$. Let $\{U_{ij} : j \in Ch(i)\}$ be a set of i.i.d. random variables independent of $\{(\beta_{ij} p_{ij}^{(r)}, \alpha_{ij}/\beta_{ij}) : j \in Ch(i)\}$. Define*

$$U_{ij}^* = \frac{U_{ij} \sum_{k \in Ch(i)} \beta_{ik} p_{ik}^{(r)}}{\sum_{k \in Ch(i)} \beta_{ik} p_{ik}^{(r)} U_{ik}}$$

*such that $\{U_{ij}^* : j \in Ch(i)\}$ are identically distributed random variables satisfying*

$$\sum_{j \in Ch(i)} \beta_{ij} p_{ij}^{(r)} U_{ij}^* = \sum_{j \in Ch(i)} \beta_{ij} p_{ij}^{(r)}. \tag{7}$$

*If there exist an experimental design $T_E^*$ such that $\mathbf{W}_i(T_E^*) = \mathbf{W}_i(T_E^{(r)})$ and $p_{ij}^* = p_{ij}^{(r)} U_{ij}^*$, then $\mathbb{E}[Y_i(T_E^*)] = \mathbb{E}[Y_i(T_E^{(r)})]$.*

*Proof.* Since $g$ is differentiable with respect to the first coordinate, and $\mathbf{W}_i(T_E^*) = \mathbf{W}_i(T_E^{(r)})$, by applying mean value theorem, we obtain

$$\mathbb{E}[Y_i(T_E^*)] - \mathbb{E}[Y_i(T_E^{(r)})]$$

$$= \mathbb{E}\left[ g\left( \sum_{j \in Ch(i)} \alpha_{ij} p_{ij}^{(r)} U_{ij}^*, \mathbf{W}_i(T_E^{(r)}) \right) \right.$$

$$\left. - g\left( \sum_{j \in Ch(i)} \alpha_{ij} p_{ij}^{(r)}, \mathbf{W}_i(T_E^{(r)}) \right) \right]$$

$$= \mathbb{E}\left[ g'(x_i^*, \mathbf{W}_i(T_E)) \sum_{j \in Ch(i)} \alpha_{ij} p_{ij}^{(r)} (U_{ij}^* - 1) \right],$$

where $g'$ denotes the partial derivative of $g$ with respect to the first coordinate, and $x_i^*$ depends on $\{(\alpha_{ij}, p_{ij}^{(r)}, U_{ij}^*) : j \in Ch(i)\}$. We complete the proof by showing that the conditional expectation of

$$V_{ij}(T_E^{(r)}) := g'(x_i^*, \mathbf{W}_i(T_E))(U_{ij}^* - 1)$$

given $\mathbf{Z}_i(T_E^{(r)}) := \{\alpha_{ij} p_{ij}^{(r)} : j \in Ch(i)\}$ is zero.

From $\sum_{j \in Ch(i)} \beta_{ij} p_{ij}^{(r)} U_{ij}^* = \sum_{j \in Ch(i)} \beta_{ij} p_{ij}^{(r)}$, we have

$$\sum_{j \in Ch(i)} \beta_{ij} p_{ij}^{(T_E)} V_{ij}(T_E) = 0. \tag{8}$$

It follows from the definition of $U_{ij}^*$ and the independence of $\{U_{ij} : j \in Ch(i)\}$ and $\mathbf{Z}_i^*(T_E) = \{(\beta_{ij} p_{ij}^{(T_E)}, \alpha_{ij}/\beta_{ij}) : j \in Ch(i)\}$ that $\{U_{ij}^* : j \in Ch(i)\}$ is a set of identically distributed random variables conditionally on $\mathbf{Z}_i^*(T_E)$. Therefore, the conditional expectation of $V_{ij}(T_E)$ given $\mathbf{Z}_i^*(T_E)$ is identical for all $j \in Ch(i)$. Thus it follows from (8) that

$$\mathbb{E}[V_{ij}(T_E) \mid \mathbf{Z}_i^*(T_E)] = 0 \text{ for all } j \in Ch(i).$$

Since $\alpha_{ij} p_{ij}^{(r)} = (\alpha_{ij}/\beta_{ij}) \times \beta_{ij} p_{ij}^{(r)}$, we have

$$\mathbb{E}[V_{ij}(T_E) \mid \mathbf{Z}_i(T_E)]$$
$$= \mathbb{E}[\mathbb{E}[V_{ij}(T_E) \mid \mathbf{Z}_i^*(T_E)] \mid \mathbf{Z}_i(T_E)]] = 0.$$

This completes the proof, since

$$\mathbb{E}[Y_i(T_E^*)] - \mathbb{E}[Y_i(T_E^{(r)})]$$
$$= \mathbb{E}\left[ \sum_{j \in Ch(i)} \alpha_{ij} p_{ij}^{(T_E)} V_{ij}(T_E) \right]$$
$$= \mathbb{E}\left[ \sum_{j \in Ch(i)} \alpha_{ij} p_{ij}^{(T_E)} \mathbb{E}[V_{ij}(T_E) \mid \mathbf{Z}_i(T_E)] \right] = 0.$$

$\square$

## A.4 Proof of Theorem 1

It follows from Assumption 2 that

$$\mathbb{E}[Y_i(T_E^*) \mid Z_i(T_E^*) = z] = \mathbb{E}[Y_i(z, \mathbf{W}_i(T_E^*))],$$

where $\mathbf{W}_i(T_E^*) = \{p_{ki}^* : k \in Pa(i)\}$. By design, $\mathbf{W}_i(T_E^*) = \mathbf{W}_i(T_E^{(r)})$ for all $i \in \Omega_r$. Therefore, for $i \in \Omega_r$,

$$\mathbb{E}\left[ Y_i(T_E^*) \frac{f_r(Z_i(T_E^*))}{f_r^*(Z_i(T_E^*))} \right]$$
$$= \int \mathbb{E}\left[ Y_i(z, \mathbf{W}_i(T_E^{(r)})) \frac{f_r(z)}{f_r^*(z)} \mid Z_i(T_E^*) = z \right] f_r^*(z) \, dz$$
$$= \int \mathbb{E}\left[ Y_i(z, \mathbf{W}_i(T_E^{(r)})) \right] f_r(z) \, dz$$
$$= \mathbb{E}[Y_i(T_E^{(r)})] = \tau_r.$$

This completes the proof, since we have

$$\mathbb{E}[\hat{\tau}_r] = \frac{1}{\Omega_r} \sum_{i \in \Omega_r} \mathbb{E}[Y_i(T_E^{(r)})] = \tau_r.$$

$\square$

## A.5 Density Estimation

Since we observe $\{Z_i(T_E^*) : i \in \Omega_r\}$, any parametric or non-parametric density estimation method can be applied for estimating $f_r^*$. For estimating $f_r$, we make the following additional assumptions:

**Assumption 3.** *Assume that $f_r(z) \propto f_r^*((\sigma_r^*(z + \mu_r^*) - \mu_r)/\sigma_r)$ for all $r \in [m]$, where $\mu_r = \mathbb{E}\, Z_i(T_E^{(r)})$, $\mu_r^* = \mathbb{E}\, Z_i(T_E^*)$, $\sigma_r^2 = \mathrm{Var}[Z_i(T_E^{(r)})]$ and $\sigma_r^{*2} = \mathrm{Var}[Z_i(T_E^*)]$.*

Assumption 3 states that the functional form of the densities of $Z_i(T_E^*)$ and $Z_i(T_E^{(r)})$ are identical, except for the mean and the variance. This allows us to obtain an estimate of $f_r$ by adjusting the mean and the variance of an estimated $f_r^*$. In Lemma 2, we provide consistent estimators of the first two moments of $Z_i(T_E^{(r)})$ from the data $\{Z_{ij}(T_E^{(r)}) : i \in \Omega_r, \; j \in Ch(i) \cap (\Omega_r \cup \Lambda_r)\}$, where $\Omega_r$ and $\Lambda_r$ are as in Algorithm 1. Finally, note that Assumption 3 is not a theoretical necessity, but a practical recommendation for avoiding overfitting in estimating the target density.

**Lemma 2.** *Assume that*

$$\rho_i := \frac{|Ch(i) \cap (\Omega_r \cup \Lambda_r)|}{|Ch(i)|} \quad and \quad \rho_i' := \frac{(|Ch(i) \cap (\Omega_r \cup \Lambda_r)| - 1)|}{(|Ch(i)| - 1)}$$

*are independent of* $|Ch(i)|$, $\mu_{r|i} := \mathbb{E}[Z_i(T_E^{(r)}) \mid i]$ *and* $\sigma_{r|i}^2 := \mathrm{Var}[Z_i(T_E^{(r)}) \mid i]$. *We define*

$$V_1^{(r)} := \sum_{i \in \Omega_r} \sum_{j \in Ch(i) \cap (\Omega_r \cup \Lambda_r)} Z_{ij}(T_E^{(r)}),$$

$$V_2^{(r)} := \sum_{i \in \Omega_r} \sum_{j \in Ch(i) \cap (\Omega_r \cup \Lambda_r)} Z_{ij}(T_E^{(r)})^2 \ \ and$$

$$V_3^{(r)} := \sum_{i \in \Omega_r} \left( \sum_{j \in Ch(i) \cap (\Omega_r \cup \Lambda_r)} Z_{ij}(T_E^{(r)}) \right)^2.$$

*Then*

$$\frac{V_1^{(r)}}{\sum_{i \in \Omega_r} \rho_i} \xrightarrow{a.s} \mathbb{E}[Z_i(T_E^{(r)})] \ \ and \ \ \frac{V_2^{(r)}}{\sum_{i \in \Omega_r} \rho_i} + \frac{V_3^{(r)} - V_2^{(r)}}{\sum_{i \in \Omega_r} \rho_i \rho_i'} \xrightarrow{a.s} \mathbb{E}[Z_i(T_E^{(r)})^2].$$

*Proof.* Let $d_i = |Ch(i)|$ denote the out-degree of member $i$. Then the expected value of $Z_i(T_E^{(r)})$ is given by

$$\mathbb{E}[Z_i(T_E^{(r)})] = \lim_{|\Omega| \to \infty} \frac{1}{|\Omega|} \sum_{i \in \Omega} d_i \mu_{r|i} = \mathbb{E}[d_i \mu_{r|i}]. \tag{9}$$

Therefore,

$$\lim_{|\Omega_r| \to \infty} \frac{V_1^{(r)}}{\sum_{i \in \Omega_r} \rho_i} = \lim_{|\Omega_r| \to \infty} \frac{\frac{1}{|\Omega_r|} \sum_{i \in \Omega_r} \rho_i d_i \mu_{r|i}}{\frac{1}{|\Omega_r|} \sum_{i \in \Omega_r} \rho_i}$$

$$= \frac{\mathbb{E}[\rho_i d_i \mu_{r|i}]}{\mathbb{E}[\rho_i]} = \mathbb{E}[Z_i(T_E^{(r)})]$$

where the last equality follows from (9) and the fact that $\rho_i$ and $\mu_{r|i} d_i$ are independently distributed.

Next, it follows from similar calculations that

$$\mathbb{E}[Z_i(T_E^{(r)})^2] = \mathbb{E}[d_i \sigma_{r|i}^2 + d_i^2 \mu_{r|i}^2]$$

$$\lim_{|\Omega_r| \to \infty} \frac{1}{|\Omega_r|} V_2^{(r)} = \mathbb{E}[\rho_i d_i \sigma_{r|i}^2 + \rho_i d_i \mu_{r|i}^2], \ \ and$$

$$\lim_{|\Omega_r| \to \infty} \frac{1}{|\Omega_r|} V_3^{(r)} = \mathbb{E}[\rho_i d_i \sigma_{r|i}^2 + \rho_i^2 d_i^2 \mu_{r|i}^2]$$

Furthermore, it is easy to verify that $\rho_i^2 d_i^2 - \rho_i d_i = \rho_i \rho_i'(d_i^2 - d_i)$. Therefore,

$$\lim_{|\Omega_r| \to \infty} \frac{1}{|\Omega_r|} [V_3^{(r)} - V_2^{(r)}] = \mathbb{E}[\rho_i \rho_i'(d_i^2 - d_i) \mu_{r|i}^2].$$

Using the independence of $(\rho_i, \rho_i')$ and $(d_i, \mu_{r|i}, \sigma_{r|i})$, we obtain

$$\lim_{|\Omega_r| \to \infty} \frac{\frac{1}{|\Omega_r|} V_3^{(r)} - V_2^{(r)}}{\frac{1}{|\Omega_r|} \sum_{i \in \Omega_r} \rho_i \rho_i'} = \mathbb{E}[d_i^2 \mu_{r|i}^2 - d_i \mu_{r|i}^2] \ and$$

$$\lim_{|\Omega_r| \to \infty} \frac{\frac{1}{|\Omega_r|} V_2^{(r)}}{\frac{1}{|\Omega_r|} \sum_{i \in \Omega_r} \rho_i} = \mathbb{E}[d_i \sigma_{r|i}^2 + d_i \mu_{r|i}^2].$$

Hence, we have,

$$\lim_{|\Omega_r| \to \infty} \frac{V_2^{(r)}}{\sum_{i \in \Omega_r} \rho_i} + \frac{V_3^{(r)} - V_2^{(r)}}{\sum_{i \in \Omega_r} \rho_i \rho_i'}$$

$$= \mathbb{E}[d_i \sigma_{r|i}^2 + d_i \mu_{r|i}^2] + \mathbb{E}[d_i^2 \mu_{r|i}^2 - d_i \mu_{r|i}^2]$$

$$= \mathbb{E}[d_i \sigma_{r|i}^2 + d_i^2 \mu_{r|i}^2] = \mathbb{E}[Z_i(T_E^{(r)})^2].$$

$\square$

## A.6 Bootstrap Variance Estimation

We draw $B$ random samples with replacement $\{\Omega'^{(1)}, \ldots, \Omega'^{(B)}\}$ of size $|\Omega'|$ from $\Omega'$. For each $t \in \{1, \ldots, B\}$ and $r = 0, \ldots, m$, we obtain $\Omega_r^{(t)}$ by selecting all elements of $\Omega'^{(t)}$ that are in $\Omega_r$. Then we obtain $\hat{\tau}_r^{(t)}$ by applying the density estimation and the estimate computation of Algorithm 2 with $\Omega_r^{(t)}$ instead of $\Omega_r$. Finally, we obtain the bootstrap variance $\hat{\sigma}^2(T_E^{(r)})$ by computing the sample variance of $\hat{\tau}_r^{(1)}, \ldots, \hat{\tau}_r^{(B)}$. Finally, note that the bootstrap parameter $B$ should be large enough to have a stable variance estimator. We found B=1000 to be good enough for our simulations and real-world experiments.

## A.7 Simulation Setup Details

### A.7.1 Graph Construction

For our simulation, we want a graph composed of a set of pre-determined clusters with significantly high intra-cluster connectivity, and sparse inter-cluster connectivity. To that end, we first obtain an undirected graph $\mathcal{G}_{BA}^{undir} = (\Omega, E_{BA}^{undir})$ by combining 10 randomly generated graphs with 5000 vertices and average degree equals $\bar{d}_{BA}$, where each graph is generated according to the Barabasi-Albert model [7] with the power of the preferential attachment equals 0.25. Then we generate an Erdös-Rényi graph [12] $\mathcal{G}_{ER}^{undir} = (\Omega, E_{ER}^{undir})$ with 50000 vertices and average degree equals $\bar{d}_{ER}$, where all pairs of nodes have an equal probability of being connected. We obtain $\mathcal{G}^{undir} = (\Omega, E_{BA}^{undir} \cup E_{ER}^{undir})$. Finally, we transform $\mathcal{G}^{undir}$ to a directed graph $\mathcal{G}$ by replacing each undirected edge $i - j$ by two directed edges $i \to j$ and $j \to i$. Note that the average degree (of the incoming edges) in $\mathcal{G}$ is given by $\bar{d}_{\mathcal{G}} := \frac{1}{|\Omega|} \sum_{i \in \Omega} |Pa(i)| = \bar{d}_{BA} + \bar{d}_{ER}$.

For the simulation study, we choose the following three settings:

| $d_{BA}$ | $d_{ER}$ | $\bar{d}_{\mathcal{G}}$ | $\bar{d}_{ER}/\bar{d}_{\mathcal{G}}$ | Sparsity | Impurity (inter-cluster edge proportion) |
|---|---|---|---|---|---|
| 20 | 1 | 21 | 1/20 | High | Low |
| 50 | 5 | 55 | 1/10 | Medium | Medium |
| 80 | 40 | 120 | 1/3 | Low | High |

**Table 2:** Simulation Settings.

### A.7.2 Cluster-based method (CB)

We take advantage of the presence of the disjoint connected components in $\mathcal{G}_{BA}$ to define a "oracle" cluster-based method. More precisely, we randomly choose two clusters to assign them $T_E^{(0)}$ and $T_E^{(1)}$ by defining

$$p_{ij}^* := \begin{cases} p_{ij}^{(r)} & \text{if } i \in Pa(j), \ j \in \mathcal{H}_r, \ r = 0, 1 \\ p_{ij}^{base} & \text{if } i \in Pa(j), \ j \in \Omega \setminus (\mathcal{H}_0 \cup \mathcal{H}_1). \end{cases}$$

where $\mathcal{H}_0$ and $\mathcal{H}_1$ are randomly chosen disjoint connected components of $\mathcal{G}_{BA}$.

### A.7.3 Data Generation Mechanism

For each directed edge $(i, j)$ in $\mathcal{G}$, we independently generate $U_{ij}$ and $V_{ij}$ from $Uniform[10, 100]$ and $Uniform[1, 2]$ respectively. We define $\alpha_{ij} = U_{ij}/d_j$ and $p_{ij}^{base} = V_{ij}/\sum_{i \in Pa(j)} V_{ij}$, where $d_j = |Pa(j)|$ denotes the degree of node $j$. We define $T_E^{(0)} = \{p_{ij}^{(0)} : (i,j) \in E\}$ and $T_E^{(1)} = \{p_{ij}^{(1)} : (i,j) \in E\}$ where $p_{ij}^{(0)} = p_{ij}^{base}$ and $p_{ij}^{(1)} \propto p_{ij}^{base} \times \left(\frac{\alpha_{ij}}{\log(1+d_i d_j)}\right)^{1/2}$, where $d_i$ is the degree of member $i$. Finally, we define

$$W_i^{(r)} = \frac{1}{d_i} \sum_{k \in Pa(i)} \alpha_{ki} \, p_{ki}^{(r)}, \quad Z_i^{(r)}(\delta) = \sum_{j \in Ch(i)} \alpha_{ij} \, (p_{ij}^{(r)})^\delta \ \text{ and}$$

$$Y_i(T_E^{(r)}, \delta) = g(W_i^{(r)} + Z_i^{(r)}(\delta)(1 + W_i^{(r)})) + \epsilon_i$$

where $g(x) = 10/(1 + \exp(-x/10))$ and $\epsilon_i \overset{\text{i.i.d}}{\sim} N(0, 1)$. Note that Assumption 1 is satisfied if and only if $\delta = 1$ and $\alpha_{ij}$'s are known. To verify robustness of our method, we choose $\delta \in \{1/4, 1/2, 1\}$ and we choose $\alpha_{ij} = 1$ for constructing $T_E^*$ using Algorithm 1.

To match with sample size of the cluster-based method, we choose $|\Omega_0| = |\Omega_1| = 0.1 \times |\Omega|$. Furthermore, we choose $|\Lambda_0| = |\Lambda_1| = 0.1 \times |\Omega|$. For choosing $C'$ in Algorithm 1, we set $q = 0.5$. For setting the optimization constraints, we choose $R_{\min} = 0$, $R_{\max} = 10$, $S_{\min} = 0.2$, $S_{\max} = 5$. For running the iterative QP, we set $K = 1000$ and $maxIter = 10$.

## A.8   Real-World Experimental Setup

We consider a directed network $\mathcal{G} = (\Omega, E)$ of all LinkedIn members (675+ million) with edges $i \to j$ if at least one content of member $i$ is considered by the baseline recommender system to be shown to member $j$ in the last 7 days. First, we randomly choose five disjoint sets subsets $\Omega_0, \Omega_1, \Lambda_0, \Lambda_1, \Gamma$ such that $|\Omega_0| = |\Omega_1| = 0.03 \times |\Omega|$ and $|\Lambda_0| = |\Lambda_1| = 0.04 \times |\Omega|$ and $|\Gamma| = 0.06 \times \Omega$. Next, we choose $C' = \Gamma \cap (\cup_{i \in \Omega_0 \cup \Omega_1} Ch(i))$. The reason for choosing $C'$ in a sightly different manner is to make $C'$ well-defined even if the underlying network is dynamic over the experiment time window.

### A.8.1   Boost factors $b_{ij}$

It follows from a straightforward calculation that the following definition of $b_{ij}$'s for $j \in C'$ will satisfy (3):

$$
b_{ij} := \begin{cases}
\dfrac{p_{ij}^* \, (1 - \sum_{k \in \Omega' \cap Pa(j)} p_{kj}^{(0)})}{p_{ij}^{(0)} \, (1 - \sum_{k \in \Omega' \cap Pa(j)} p_{kj}^*)} & \text{if } i \in Pa(i) \cap \Omega' \\
1 & \text{if } i \in Pa(i) \setminus \Omega'.
\end{cases}
$$