[Reviews · NeurIPS 2020]

Review 1

Summary and Contributions: The authors propose a method called "Optimal Allocation Strategy and Importance Sampling Adjustment" (OASIS) to provide a randomized testing framework for large-scale markets with network inference that violates SUTVA. The proposed method continues to work even with dense networks (where random treatment is particularly hard). The authors provide theoretical guarantees, test for robustness using simulations, and validate the method with a real-world experiment.

Strengths: This research tackles and important problem (how to experiment when SUTVA will certainly be violated due to network effects). The authors frame the problem nicely and propose a novel method to address it.

Weaknesses: It is not intuitive to me why OASIS would get better as the network density increases. This should be explored further.

Correctness: To the best of my knowledge, yes, but I don't understand all of it.

Clarity: Yes, it is very well-written and motivated. (Best of my batch of reviews by far.). That said, it is a very technically dense paper and I had a hard time following some parts. For optimal impact and accessibility of the work, it might be worth taking another editorial pass through for clarity, taking the time/space to explain things in greater detail and pushing more to the supplemental material.

Relation to Prior Work: This work positions itself well among related research and, to the best of my knowledge, is missing no critical citations.

Reproducibility: Yes

Additional Feedback: The notion of a shadow population is interesting, but I'm not sure how this differs from a control group (or if it does?). == UPDATE, because it was universally requested == I still think this a good paper. :)


Review 2

Summary and Contributions: A two-step method for the average treatment effect estimation in a large-scale network is proposed.

Strengths: (1) This is a significant and valuable problem in data analytics (especially for network-based topics, e.g., social networking computing). (2) The design of the method is based on sufficient theoretical analysis and solid experiments. (3) The method has a strength in its generality in handling different tasks. (4) Applause for making the code accessible.

Weaknesses: (1) Related work should be placed in a separate session. (2) To emphasize the strength of the model in benefiting different research areas, more discussion regarding the applications of the proposed method should be added to the related work and discussion. The method may be used in many social network computing topics, say influence propagation model, graphic modeling for stock connection networks, etc. The papers below may be useful. Xiao et al. "Price shock detection with an influence-based model of social attention." ACM Transactions on Management Information Systems (TMIS) 9.1 (2018). Liu et al. "An influence propagation view of pagerank." ACM Transactions on Knowledge Discovery from Data (TKDD) 11.3 (2017).

Correctness: Both look fine.

Clarity: The paper is well written.

Relation to Prior Work: Yes.

Reproducibility: Yes

Additional Feedback:


Review 3

Summary and Contributions: The aim of this paper is to propose a randomized testing framework, called OASIS, for large-scale marketplace problems in the presence of interference. There are three contributions. The first one is an approximate randomized controlled experiment. The second one is an importance sampling adjustment to correct for any leftover bias. The third one include theoretical guarantees, simulations, and a real-world experiment.

Strengths: This is a well-written paper with some interesting ideas. There are three contributions. The first one is an approximate randomized controlled experiment. The second one is an importance sampling adjustment to correct for any leftover bias. The third one include theoretical guarantees, simulations, and a real-world experiment.

Weaknesses: 1. How do you calculate $Y_i(T_E^{(r)})$ for a single member given Pa(i) and Ch(i)? Are p_{ij}^{(r)} and p_{ij}^{base} given or calculated from real data? How to do it? 2 Please check line 152. You have {\bf Pa}(i). Have you ever define it? 3. The proposed method depends on the existence of a set of "intervening variables". My question would be how to ensure that such existence is a reasonable one. Is there a way to check these two assumptions in practice. 4. In Line 166, where is $\Omage_r$ in Figure 1b? Please clarify. 5. In line 5 of Algorithm 1, what is (4)? Please clarify. 6. The authors claimed that the bootstrap method works in this case. Is there any theoretical result to ensure this?

Correctness: The claims and method are correct.

Clarity: This paper is well written in english. The motivation underlying the procedure is not well explain. It would be better to present a path-diagram for the whole procedure.

Relation to Prior Work: Yes.

Reproducibility: Yes

Additional Feedback:


Review 4

Summary and Contributions: The authors look at running experiments on producer-consumer recommendation platforms. They set-up assumptions which determine the structure of potential outcomes, under which SUTVA is violated and describe am edge-level randomized design that allows for unbiased estimation (by matching producer and consumer-side probabilities). They validate their results on simulated data, and discuss a real world example.

Strengths: The paper tackles an important and particularly thorny topic, that of interference, in one of the toughest settings: recommendation systems between users that act both as producers and consumers of content. The authors do a good job of distilling the problem down to some useful notation, and assumptions. They prove the feasibility of their modeling assumptions by using this framework in a live experiment.

Weaknesses: Three weaknesses that are not dealbreakers but could improve the paper: - the linear exposure assumption that is discussed, and in general the model of potential outcomes, has ties to other recent works on exposure mappings (savje), exposure assumptions in bipartite experiments works (papadogeorgou and others). The authors could perhaps be a little more thorough in their related works section. - the live experiment does demonstrate the feasibility of this approach, but doesn't provide much insight into how it improved over the baseline of clustering. Is there evidence that these results were less biased than we would have expected otherwise? - the writing at times was a little clunky and could've been improved by mixing less mathematical notation into english sentences. Perhaps consider making the notation a little lighter and using more sentences not interspersed with mathematical notation which can be difficult to read (esp. section 3 and the description of the real-world experiment).

Correctness: I believe so.

Clarity: Yes. Some notation could be made a little lighter.

Relation to Prior Work: Yes, though I believe the related works section could be a bit more comprehensive in its review of the literature.

Reproducibility: Yes

Additional Feedback:

[Author Response · NeurIPS 2020]

We thank all reviewers for appreciating the importance and the challenge of the problem we tackle in this paper. We are glad to hear that all reviewers found the paper to be well written.

**Reviewer 1**:

1. With increasing density in the graph, the experiment design part will have more flexibility to redistribute the exposure optimally, and the post-experiment adjustment part will have a larger sample size for density estimation (see Section A.5 in the supplement). We will add these to the paper.

2. The shadow population is not the same as the control population. The shadow population members are randomly assigned to consumer-side treatment or consumer-side control (as in classical A/B testing design). In the absence of interference (i.e., the classical setting), the shadow population is the same as the measurement population. However, these two populations are different in the presence of interference (i.e., non-zero producer-side effect), since the shadow population members do not get the correct producer-side experience.

**Reviewer 2**:
Thank you for sharing the papers. We will cite and discuss the relevant ones.

**Reviewer 3**:

1. In Theorem 1, we do not use $Y_i(T_E^{(r)})$ to compute the average treatment effect (since they are unobservable). Instead, we apply importance sampling adjustment to the observed response $Y_i(T_E^*)$. The importance weights are computed based on the observed $Z_i(T_E^*)$ values and estimated densities of $Z_i(T_E^*)$ and $Z_i(T_E^{(r)})$. For the density estimation of $Z_i(T_E^{(r)})$ from the experimental data, please see Section A.5 in the supplement. Note that we do not need to know the $p_{ij}$ values at this point. However, we use the $p_{ij}^{(r)}$ and the $p_{ij}^{base}$ values for the design of experiment (see eq. 2). To this end, we assume that the $p_{ij}^{(r)}$ and the $p_{ij}^{base}$ values are known in the simulation settings, whereas we estimated them from the data in the real-world experiment (lines 285–293).

2. Thanks for catching this. It should be $Pa(i)$ (see line 105).

3. The presence of an intervening variable is indeed a strong assumption. However, it is a natural assumption in some situations. For example, in the content market place setting, it is a reasonable assumption that the producer-side experience of a member is driven by the total feedback received from the content-consumers. It is challenging to develop a sound method for verifying this assumption, and it is beyond the scope of this paper.

4. Thanks for catching the typo. It should be Figure 2 instead.

5. Thanks again, it should be eq. (2).

6. Note that the estimator in Theorem 1 is an empirical average when the densities are known, implying the consistency of the bootstrap variance estimators under mild assumptions. When the densities are unknown, the consistency of the bootstrap variance relies on the consistency of the density estimation (which holds under some regularity conditions). We will add references.

**Reviewer 4**:

1. Thank you for sharing the papers. We will cite and discuss the relevant ones.

2. In our real-world experiment, we need to have millions of observations to detect small improvements in metrics (1-2%). While OASIS experiments can have millions of observations, the state-of-the-art ego-cluster experiments would be, for example, limited to 100K i.i.d. observations if we fix the proportion of inter-cluster and intra-cluster edges to 1:4. In this particular setting, we do not have a reasonable cluster-based baseline to compare with. However, this is indeed a valuable future work to compare OASIS with a reasonable baseline in real-world experiments.

3. We appreciate your feedback, and we will try our best to improve the readability.

[Meta-Review · NeurIPS 2020]

This paper is concerned with AB Testing in real-world settings where the common SUTVA assumption does not hold: in the context of Internet marketplaces for instance, treatments applied to different producers might have effects on other producers as all producers are related via the consumers. The proposed approach considers the case where the treatment consists of modifying the probability of showing a content of producer i to consumer j. The proposed approach combines several steps: i) assuming an intervening variable (mediator) capturing the global effect of a treatment, ii) defining auxiliary variables attached to each edge (ij) to estimate their impact w.r.t. the mediator; iii) showing that these auxiliary variables are solutions of a quadratic problem optimization ; iv) defining a relaxation thereof to achieve scalability and studying the convergence error. The reviewers particularly appreciated the relevance of the problem, the generality of the approach, and the fact that the code is accessible. The AC congratulates the authors for this nice piece of work.